# A Review of Ketogenic Dietary Therapies for Epilepsy and Neurological Diseases: A Proposal to Implement an Adapted Model to Include Healthy Mediterranean Products

**DOI:** 10.3390/foods12091743

**Published:** 2023-04-22

**Authors:** Cinzia Ferraris, Monica Guglielmetti, Lenycia de Cassya Lopes Neri, Sabika Allehdan, Jamila Mohammed Mohsin Albasara, Hajar Hussain Fareed Alawadhi, Claudia Trentani, Simone Perna, Anna Tagliabue

**Affiliations:** 1Ketogenic Metabolic Therapy Laboratory, Department of Public Health, Experimental and Forensics Medicine, University of Pavia, 27100 Pavia, Italy; cinzia.ferraris@unipv.it (C.F.);; 2Laboratory of Food Education and Sport Nutrition, Department of Public Health, Experimental and Forensic Medicine, University of Pavia, 27100 Pavia, Italy; 3Department of Biology, College of Science, University of Bahrain, Sakhir P.O. Box 32038, Bahrain; 4Division of Human Nutrition, Department of Food, Environmental and Nutritional Sciences (DeFENS), University of Milan, 20122 Milano, Italy

**Keywords:** ketogenic dietary therapies, Mediterranean diet, practical recommendations, neurological disease, cardiovascular risk

## Abstract

Based on the growing evidence of the therapeutic role of high-fat ketogenic dietary therapies (KDTs) for neurological diseases and on the protective effect of the Mediterranean diet (MD), it could be important to delineate a Mediterranean version of KDTs in order to maintain a high ketogenic ratio, and thus avoid side effects, especially in patients requiring long-term treatment. This narrative review aims to explore the existing literature on this topic and to elaborate recommendations for a Mediterranean version of the KDTs. It presents practical suggestions based on MD principles, which consist of key elements for the selection of foods (both from quantitative and qualitative prospective), and indications of the relative proportions and consumption frequency of the main food groups that constitute the Mediterranean version of the KDTs. We suggest the adoption of a Mediterranean version of ketogenic diets in order to benefit from the multiple protective effects of the MD. This translates to: (i) a preferential use of olive oil and vegetable fat sources in general; (ii) the limitation of foods rich in saturated fatty acids; (iii) the encouragement of high biological value protein sources; (iv) inserting fruit and vegetables at every meal possible, varying their choices according to seasonality.

## 1. Introduction

Neurological diseases have complex pathogenesis and, for many of them, effective forms of treatment are still under development. Conventional therapy is often associated with increased tolerance and/or drug resistance [1]. Consequently, more effective therapeutic strategies are being sought to increase the effectiveness of available forms of therapy and improve the quality of life of patients. Among dietary interventions, ketogenic dietary therapies (KDTs) can provide potential therapeutic benefits in patients with neurological problems by effectively controlling the balance between pro- and antioxidant processes and pro-excitatory and inhibitory neurotransmitters, and modulating inflammation or changing the composition of the gut microbiome [1]. Dietary ketosis has been demonstrated to be beneficial for drug-resistant epilepsy [2,3,4,5] and there is accumulating evidence for several other neurological diseases, specifically migraines [6,7], brain tumors [8,9,10], Alzheimer’s disease [11,12], Parkinson disease [13,14], and autism spectrum disorder [15,16]. After the demonstration of seizure reduction following a fasting treatment in 1921, KDTs have been developed, mimicking the biochemical changes of fasting without malnutrition (i.e., isocaloric ketogenic diets where fatty foods substitute for carbohydrates to maintain an adequate energy intake and avoid weight loss). Nowadays, various KDTs are available and used in the treatment of neurological disorders [17]. We can distinguish the Classical Ketogenic Diet (CKD), Medium Chain Triglycerides ketogenic diet, (MCT–KD), the Modified Atkins Diet (MAD), and the Low Glycemic Index Treatment (LGIT). Based on the growing evidence of the therapeutic role of KDTs for neurological diseases and on the protective effect of the MD, it would be of great importance to delineate a Mediterranean version of this high-fat therapy in order to maintain its high ketogenic ratio and avoid side effects, especially in patients requiring long-term treatment (i.e., GLUT1-DS patients).

This proposal aims to provide the key elements for the selection of foods, both in a quantitative and qualitative perspective, indicating the relative proportions and consumption frequency of intake of the main food groups that constitute the Mediterranean version of the KDTs (Med–KDTs).

## 2. Materials and Methods

The present narrative review was performed according to the following steps:Configuration of a working group: six operators skilled in clinical nutrition with one acting as a methodological operator and four participating as clinical operators;Formulation of the revision question on the basis of considerations made in the abstract: “Ketogenic dietary therapies for epilepsy and neurological diseases: A proposal to implement an adapted model to include healthier Mediterranean products”;Identification of relevant studies for our purpose, performed as follows:Research strategy was planned on PubMed and Scopus based on the definition of the keywords of the interest field of the documents to be searched, grouped in inverted commas (“…”), and used separately or in combination;Use of the Boolean AND operator, which allows for the establishment of logical relationships among concepts;Research modalities: advanced search;Limits: papers published until January 2023; humans, animals, in vivo, and in vitro studies; languages: English;Manual search performed by senior researchers experienced in clinical nutrition through the revision of reviews and research articles on “Ketogenic diet in neurological diseases and Mediterranean Diet” published in qualified journals of the Index Medicus.


## 3. Ketogenic Dietary Therapies

The common features of the KDTs are very high-fat, low carbohydrates, and moderate protein content, even if they differ in the relative macronutrient contribution and ketogenic ratio (KR). The ketogenic ratio refers to the ratio of grams of fat to grams of protein plus carbohydrates combined. The most common ratio is 4 g of fat to 1 g of protein plus carbohydrate (described as “4:1”); 90% of calories are from fat [17], but in the CKD it can range from 1:1 to 4:1, according to the individual therapeutic needs. It can also be applied to the other KDTs when data on macronutrient composition are available.

A detailed description of the macronutrient composition of these KDTs and their respective KR is represented in Table 1.

The CKD [18] was the first protocol developed in the 1920s to mimic the biochemical changes of fasting for the treatment of drug-resistant epilepsy (DRE). Since then, it has become an important non-pharmacological therapy for this condition [2]. The CKD also represents the mainstay of treatment for glucose transporter type 1 deficiency syndrome (GLUT1-DS, OMIM 606777) [21], which is currently the only available treatment and should be maintained for life [22]. It is also the first choice treatment for pyruvate dehydrogenase deficiency (PDHD), in which pyruvate cannot be metabolized into acetyl-CoA, resulting in a mitochondrial disorder with lactic acidosis, seizures, and severe encephalopathy [23]. In both disorders, KDTs provide ketones that bypass the metabolic defects and serve as an alternative cerebral fuel for the developing brain.

The MCT–KD was introduced by Huttenlocher in 1971 as a more liberal option to the CKD. Fats are reduced to 70% of the total daily energy and are composed primarily of medium-chain triglycerides (MCTs) [19]. The remaining part derives from long-chain triglycerides (LCTs). The MCTs quote is about 10–25% in infants and increases to 40–60% for individuals aged > 2 years [24,25]. MCTs do not require digestion and carnitine use but are carried directly to the liver for ketone body production. This characteristic allows MCTs to increase ketone bodies production, permitting the consumption of a higher intake of carbohydrates and proteins and lower fat intake. MCTs can be found in different products, such as coconut oil, MCT oil, creams, etc. In order to reduce the possible risk of gastrointestinal adverse effects (i.e., vomiting, diarrhea, abdominal pain, etc.), linked to MCTs consumption, it is recommended to equally distribute MCTs at every meal (up to six times per day) and to gradually increase the total amount [17,19,26].

The Modified Atkins Diet (MAD) was first proposed by Kossoff in 2003, utilizing the induction phase of the original Atkins diet [4]. Carbohydrates are limited to 10 g/day in children, 15 g/day in adolescents, and 20 g/day in adults. Fatty foods are encouraged to cover energy requirements, ideally reaching about 65% of calories. Although protein is not restricted, specific portion sizes of protein-rich foods are suggested in order to not impact the ketosis levels through gluconeogenesis. In fact, intakes above the needs of the average adult (0.8 to 1.2 g/kg actual or adjusted weight for an adult) or above the dietary reference intake for age in pediatrics and adolescents may negatively affect ketosis [27].

The LGIT was coined by Pfeifer and Thiele in 2005 [20]. It allows carbohydrate intake of up to 40–60 g/day but includes only carb-rich foods with a glycemic index of less than 50 in order to minimize glycemic fluctuations, specifically at the brain level [28]. Protein and fat intakes are monitored, but not so strictly as in the classical protocol. All these protocols are characterized by a very high fat content (ranging from 60 to 90% of energy) which is mandatory both to achieve a high ketosis level and to reach the energy requirements of the patients, avoiding weight loss. Therefore, KDT administration requires constant careful dietary planning and nutritional monitoring over time, either to ensure its effectiveness or to reduce the consequences of marginal or overt nutritional deficits (energy, proteins, minerals, and vitamins) or of nutrient excess (lipids, saturated fat, and cholesterol) that are peculiar to these diets. The need to dramatically reduce carbohydrate intake, in fact, results in the necessity to limit the assumption of fruits and vegetables, raw cereals, and/or enriched grains. Thus, the vitamin, mineral, and fiber contents of these diets are unbalanced and inadequate to reach the specific recommendations for age and sex [29,30]. The international guidelines for the management of KDTs [17] highlight the importance of adequate multivitamin–mineral supplementation through carbohydrate-free or minimal carbohydrate-containing products. Fiber-containing products can be added if constipation occurs. Carnitine supplementation is recommended only if either level is low (due to prolonged use of anticonvulsant drugs, poor nutritional status, or the long-term use of KDTs) or children become symptomatic (generalized weakness, excessive fatigue, and decreased muscle strength) [17]. On the other hand, the achievement and maintenance of energy requirements based on fatty foods determine a significant increase in the percentage of energy derived from saturated fatty acids (SFA) and unsaturated fats (monounsaturated or MUFA and polyunsaturated or PUFA) in comparison to baseline dietary intakes. When considering a typical daily plan designed for children with drug-resistant epilepsy, the percentages change from baseline as follows: from 10 to 21% for SFA, 13 to 27% for MUFA, and 3 to 9% for PUFA [31]. From a theoretical point of view, what matters to maintain KR is the amount of fat to consume. However, from the nutritional point of view, dietary fatty acid composition is also important. Paying attention to food quality (i.e., increasing the ratio of PUFA + MUFA versus SFA) is essential to reduce the risk of hyperlipidemia in patients on long-term KDTs [32,33,34,35].

## 4. Lipid Profile and Cardiovascular Risk during Long-Term Use of KDTs

Although the benefits of KDTs in the control of epilepsy and other neurological diseases are well documented in the literature, several adverse effects have been described during long-term KDT such as hyperuricemia, hyperlipidemia, kidney stones, and delayed growth in children [17]. Among these effects, the most common clinical manifestation observed in the studies is dyslipidemia, according to Chesney et al. [36], Liu et al. [37], Kwiterovich et al. [38], Nizamuddin et al. [39], and Perna et al. [40]. These studies showed that the increase in total cholesterol (CHOL-t), low-density lipoprotein (LDL), as well as triacylglycerol (TG), were the main adverse effects on lipid metabolism observed throughout KDT intervention. Hyperlipidemia is observed in all kinds of KDTs and is often temporary [17] and may be partially reversed by dietary modifications such as the consumption of MCT and olive oil and decreasing the consumption of cholesterol, trans, and SFA (all fatty meats, egg yolk, heavy cream, butter, animal fat, palm oil, and coconut oil), supplementing with omega-3 fatty acid or carnitine, decreasing the KR, or using solely formula-based KDTs [37,39]. Recently Yılmaz, et al. (2022) showed an increase in CHOL-t and TG concentrations at the beginning of KDT (especially in the first month), which persisted for 24 months. These authors managed to counter hyperlipidemia using dietary interventions such as replacing SFA with unsaturated fats or MCTs, carnitine and/or omega-3 fatty acid supplementation, or reducing the KR gradually down to the level of 2:1, thus trying not to alter ketosis and seizure control [41]. In addition to the changes in the lipid profile, previous studies have shown that oxidative modifications of lipoproteins as well as the increase in small particle LDL levels are essential for the initiation and progression of atherosclerosis in both adults and in pediatric patients [42,43,44]. Therefore, these studies indicate that the increase in cardiometabolic risk is not limited to the cholesterol concentration associated with LDL, but also the quality of dietary fats is important. There are no data from long-term prospective studies that evaluated the impact of KDTs on cardiovascular health. Even if changes in lipid profile in the long term do not seem to be significant, the influences of these changes on coronary heart disease are unknown [45]. Kapetanakis et al. [46] and Doksoz et al. [47] reported a negative impact of the CKD on the lipid profile without an effect on the elasticity and thickness properties of the carotid artery after 24 and 6 months of KDT, respectively. However, these studies did not describe the fatty acid composition of the KDT and did not monitor any marker of oxidation. Various factors might cause controversial results, for example, genetic background, duration of KDTs, food composition, quality, and sources of fats [48]. According to Grosso et al. [49], preferential consumption of plant-derived foods and dairy products is likely to exert anti-inflammatory effects. On the other hand, animal products, such as red meat and eggs, could have a potential pro-inflammatory effect, which may be suppressed by the consumption of fiber-rich foods. The strict composition of KDTs often precludes the inclusion of fiber-rich foods, which instead are particular elements of the MD. Unlike the usual Western diets, the MD includes a variety of minimally processed and plant foods that are rich in antioxidant vitamins (β -carotene and vitamins C and E), organic folate, phytochemicals (e.g., flavonoids), and minerals such as selenium. The consumption of fish, meat, eggs, and cheese, on the other hand, offers additional nutrients; in particular, vitamin B12, which is absent in a completely plant-based diet [50,51]. Moreover, it has a specific lipid profile composition, abundant in MUFA and PUFA (derived primarily from oleic acid in olive oil). The MD is also rich in other components such as polyphenols, which have antioxidant and anti-inflammatory qualities. All these features are important contributors to the protective effect of the MD against CVD. Many studies have shown its effectiveness in preventing both total mortality and CVD and their associated risk factors, such as diabetes, high cholesterol levels, and high blood pressure [52,53,54,55]. In fact, the MD is associated with better cardiovascular health outcomes, including clinically meaningful reductions in rates of coronary heart disease, ischemic stroke, and total cardiovascular disease (CVD) [56], and a reduced prevalence of metabolic or neurodegenerative diseases [57,58]. The biomolecular processes that appear to promote long-term health effects of the MD include: (a) the reduction in lipid levels, inflammatory responses, and platelet accumulation; (b) the suppression of nutrient-sensing pathways by restricting specific amino acids; (c); the synthesis of intestinal metabolites with increased protection against oxidative stress [51].

## 5. Current Studies Available on Mediterranean Ketogenic Diet

A few modifications have been proposed recently towards a “Mediterranean version” of high-fat KDTs to improve their nutritional profile. Two dietary protocols were proposed by Italian [59] and Spanish [60] authors in 2011, but they regard the implementation of low-calorie ketogenic diets for weight loss purposes, which is far from the objective of this paper, so they were excluded.

Only three studies completely met the aforementioned criteria, using a “Mediterranean version” of KDTs for neurological diseases. The first one was published by Guzel et al. in 2019 [61], who used an “olive-based CKD” in 389 children with drug-resistant epilepsy. Total daily energy was 60–80 kcal/kg, whereas 80–85% of total fat was derived from extra virgin olive oil (other fat sources were cream, meats, and butter, excluding palm oil, sunflower oil, hazelnut oil, coconut oil, or corn oil). Palm oil, sunflower oil, hazelnut oil, coconut oil, or corn oil were not used. Green vegetables and avocados were employed abundantly in patients’ daily menus to prevent and treat constipation. Protein intake was 1–1.5 g/kg/day. Non-fasting gradual initiation protocol with a KR ratio between 2.5:1 and 4:1 was used. The menus were prepared according to traditional Turkish cuisine to increase patient compliance and palatability. Patients who had a nasogastric tube or gastrostomy, or were under the age of one year and had feeding problems, received a commercial KD formula. A blood beta-hydroxybutyrate measurement was used to evaluate the ketone status with the aim to maintain the levels between 4 and 5 mmol/L. Even with this approach, hyperlipidemia was the most common new-onset adverse effect, noted in half of the patients. If a patient was diagnosed with hyperlipidemia, the diet was modified by reducing dietary fats by 20–25% without affecting blood ketone levels and by eliminating egg yolk and SFA sources (cream, butter, fatty meats) from the diet. Only three patients out of 198 did not respond to dietary modification and needed statin prescription. The addition of extra virgin olive oil was the main modification to the “Modified Mediterranean Ketogenic Diet” (MMKD), as experimented by Nagpal et al. in 2019 [62] and Neth et al. in 2020 [63] in older adults at risk for Alzheimer’s disease. Despite not being specified as the MAD, these studies used a similar target macronutrient composition (% of total calories). The diet was composed of <10% carbohydrates (less than 20 g/day), 60–65% fat (plus 2 and 1 L of olive oil, respectively), and 30–35% protein. The menus contained high levels of healthy fats and low-SFA proteins such as fish and lean meat. Fruits and vegetables were limited and low-carbohydrate store-bought products plus artificially sweetened beverages were avoided; only one glass of wine/day was allowed. Blood ketones were measured weekly to ensure compliance with the MMKD and an increase of 0.7 mmol/L versus baseline was reported after six weeks on the diet [62]. These studies compared the effects of the MMKD to the American Heart Association Diet (AHAD) on microbiota, CSF markers, neuroimaging metabolism, and cognition in moderate cognitive impairment (MCI). AHAD is a low-fat diet with <40 g/day lipids (15–20% of total caloric intake), proteins represented 20–30% of daily caloric intake, and the remaining 55–65% was derived from carbohydrates (including plentiful fruits, vegetables, and adequate fiber). Nagpahl (2019) [62] demonstrated how MMKD distinctly influences the gut microbiome and SCFAs as well as their associations with Alzheimer’s CSF biomarkers in subjects with or without MCI. Neth et al. (2020) [63] evidenced increased cerebrospinal fluid Aβ42, cerebral perfusion, and ketone body uptake (11C-acetoacetate PET, in subsample) and decreased tau following the MMKD for 6 weeks. Concerning metabolic profile, Neth et al. (2020) [63] reported a reduction in HbA1c, glucose, insulin, TG, and VLDL cholesterol in subjects with pre-diabetes after 6 w of the MMKD. All the aforementioned studies are synthesized in Table 2.

## 6. Proposal of a Mediterranean Version of KDTs for Neurological Diseases

The macronutrient composition of the Mediterranean diet and KDTs are opposite, with carbohydrates and fat-rich foods being the main energy source in the MD and in KDTs, respectively. Therefore, the quality of fatty acid composition and frequency of consumption are the main points to be considered. Taking into consideration the limited suggestions from existing literature, we elaborated practical recommendations for a Mediterranean version of the KDTs based on MD principles [64] and our long-term clinical experience [31,40,65,66,67,68,69]. Our analysis focused on the most common foods available on the market, which could suit KDTs, recently revised in Italy by Leone and colleagues [70].

### 6.1. Fat Sources

According to the MD principles [64], fats and fat-rich foods, especially those of animal origin, should be consumed in limited amounts, with the only exception being olive oil, which can have a moderate intake. On the contrary, due to the therapeutic composition of the KDTs, fats and fat-rich foods are the most representative in these protocols and naturally have a high KR, ranging from low to high, according to the calculations, independently of the fat quality.

Olive oil is the principal source of fat recommended in the Mediterranean because of its high nutritional quality [64]. The MD recommends the daily consumption of olive oil at every meal, preferably raw in order to maintain its quality [71]. It can also be cooked because it can resist high temperatures. It has a high content of monounsaturated oleic acids and an abundance of antioxidant compounds, particularly in extra virgin olive oil. These suggestions can also be maintained in KDTs as a possible strategy to prevent hyperlipidemia, as described in the international guidelines for the management of KDTs and other studies [17,72]. In order to reach the amounts of lipids prescribed in KDTs, other fat sources must be inserted into the dietary plan. Vegetable oils, for example, sunflower, wheat germ, and rice oils, can be used in addition to olive oil. They are rich in PUFA and other compounds such as, for example, γ-oryzanol, a bioactive compound with antioxidant and hypolipidemic effects present in rice oil [68]. Another oil of vegetable origin that can be very useful in KDTs is coconut oil due to its high MCT content. MCTs have a ketone-boosting effect and their replacement of long-chain fatty acids allows for increasing carbohydrate intake. However, as coconut oil is more than 90% SFA, its amount has to be evaluated in the context of other fatty foods of animal origin. Reflecting MD indications, fats of animal origin (i.e., butter, cream, mascarpone cheese, mayonnaise) should be consumed in moderation and some of them (i.e., pork lard) left for special occasions.

Olives, nuts, and seeds should be consumed daily, according to the MD, as good sources of healthy lipids, proteins, vitamins, minerals, and fiber [73]. Their composition also suits the KDTs, in which they should represent the basis of fat sources, in addition to olive oil. Several epidemiological studies and trials have shown an inverse association between nut consumption and the risk of hyperlipidemia and cardiovascular disease [74].

A further strategy to increase PUFA intake consists of the addition of soy lecithin, commercially available in the granular form, or avocado to the meal. Avocado is an optimal source of ω-6 and ω-9, fiber, vitamins, and minerals and it contains an oil rich in MUFA, which appears to enhance nutrients and phytochemical bioavailability [75,76]. It can be eaten by itself or used in cream preparation (i.e., guacamole sauce) with oil. Avocado fruit has nutrient and phytochemical profiles similar to tree nuts (almonds, pistachios, or walnuts), which have qualified heart health claims [75,77].

### 6.2. Protein Sources

Consumption of a variety of plant- and animal-origin proteins is recommended. Traditional Mediterranean dishes do not usually have animal-origin protein foods as the main ingredient but rather as a source of flavor. KDTs require moderate protein intakes in order to reach a high KR. Protein sources are usually selected from animal foods because of their high-quality protein content which permits satisfying the essential amino acid needs with moderate intakes. These foods (i.e., meat, processed meat, fish, dairy, and eggs) are also rich in animal fat, and, according to the MD, their frequency of consumption should be varied in a week.

#### 6.2.1. Animal-Origin Proteins

Fish and shellfish are a good source of high-quality protein and healthy lipids. Varied consumption is recommended; at least two times a week according to MD principles [64], with a preference for fatty fish. In particular, salmon and mackerel should be privileged due to their high EPA and DHA content, compared to other fish or high-protein foods. In fact, fish, especially those high in lipids, and shellfish consumption has been reported to reduce the risk of CVD, as they have anti-inflammatory properties due to their content of long-chain n-3 PUFA [78].

In the Mediterranean region, a wide variety of fermented dairy products is available, i.e., plain yogurt, kefir, cheese, and ricotta, which contain high bioavailable vitamins and minerals (i.e., calcium), and branched-chain amino acids (BCAA) with health-promoting effects [79]. Moreover, yogurt and kefir are beneficial to gut microbiota and, by the gut–brain axis, can modulate an individual’s overall health by preventing non-communicable diseases and metabolic disorders [80,81,82]. Although their richness in calcium is important for bone and heart health, dairy products can be an important source of SFA [83]. Reflecting the recommendations of the Mediterranean diet, dairy products must be consumed daily (at least two portions per day). Regardless of the absence of consensus about the consumption of low- or whole-fat dairy products and chronic diseases [84], in KDTs, those higher in fat should be preferred in order to achieve lipid and energy requirements. The present proposal considers it important to favor milk and yogurt daily and limit cheese to weekly consumption.

In relation to meat, we considered white, red, and processed meat. The MD favors the consumption of white meat (i.e., chicken, turkey) as a good source of protein without the high levels of SFA found in some red meat (i.e., beef, pork). However, a moderate amount of red (less than two servings) and processed meat less or equal to one time a week) is tolerated and can be maintained in KDTs. Processed meat includes raw ham, cooked ham, sausage, wurstel, bacon, and hamburger; they are often rich in salt and SFA. Among these, raw ham and ham should be consumed more frequently compared to other types of processed meat due to their HBV protein content and their lower level of processing compared to the others.

Eggs are sources of high biological value proteins, lipids, cholesterol, and water. According to MD guidelines, egg consumption, including those used in cooking as well as baking, should be 2–4 times per week. These consumption frequencies can also be used in planning KDTs in which eggs are among the frequently used foods due to their lipid content. The latest data available on the Italian population reported that high consumption of eggs (even within the recommended intake) was linked to an increased risk of all-cause mortality and CVD [85]. A substantial part of this association was likely to be determined by the contribution of eggs to dietary cholesterol. For this reason, given the high fat intake of KDTs, we recommend limiting the consumption of eggs according to the dietary prescription and to one’s own blood lipid profile.

#### 6.2.2. Legumes

Legumes (i.e., peas, beans, soybeans, chickpeas) are components of high importance in the diet of many countries of the Mediterranean region. They are considered high-protein, sustainable, very versatile, and culturally diverse foods found all over the world. The versatility of legumes covers a wide range of possibilities through their use in plant-based dairy analogs and plant-based flours, providing alternative protein and maximal amounts of nutrients and bioactive compounds [86]. In the MD, legumes are a healthy plant protein that should be considered as a meat alternative (more than two servings per week). In a study comparing a plant-based low-fat diet versus an animal-based ketogenic diet in an ad libitum energy intake environment, Hall et al. (2021) found higher glycemic load in the first diet and higher energy density in the second one [87]. Considering the high content of carbohydrates, the amount of legumes must be limited in the more restrictive KDTs (CKD) [88], whereas it can be greater in more liberal diets (MAD and LGIT).

### 6.3. Carbohydrates and Fiber Sources

In contrast with MD indications, in which carbohydrates should be the main source of daily energy, KDTs permit only small amounts of carbohydrates. Carbohydrate restriction varies in the different KDT protocols, with the CKD as the most restrictive one. The main sources of carbohydrates allowed in these diets are fruit and vegetables, low-carbohydrate products, and, in some cases, low glycemic index foods, as long as total carbohydrate intakes do not exceed the prescribed ones. Among all the foods rich in this macronutrient, fruits and vegetables should be privileged as sources of fiber, minerals, vitamins, and water, which are lacking in these diets. Fruits and vegetables should be varied and chosen according to their seasonality, following MD principles, and their sugar content. Moreover, when choosing the type of fruit to be consumed, it is important to take into account its antioxidant content. Blackberries, apples, tangerines, and lemons, for example, are rich in these protective components.

Regarding low-carb products, it is important to distinguish between high-protein low-carbohydrate food products and high-fat products. The first ones are commercialized mainly for weight loss purposes and even if defined as “ketogenic”, they actually are not because of their high protein and low fat content that reduces the ketogenic ratio and thus negatively impacts on ketosis. On the contrary, high-fat products specifically produced for KDTs include powder or liquid ketogenic formulas, medium-chain triglyceride-containing products, ketogenic baking mixes, and ready-to-eat ketogenic products [70], and can be inserted into the menu according to the prescription. Although not carbohydrate sources, we also included in this paragraph glucomannan-based products and sweeteners, often used as substitutes for carbohydrate-rich foods. Glucomannan-based products, originally developed for the dietary treatment of obesity [89], are also used in the ketogenic dietary treatment of neurological diseases [70]. They are fiber-rich products used as substitutes for classical cereal-based foods (such as pasta and rice). They are acaloric and may have potential benefits for patients treated with KDTs. This is linked to their fiber content, which can have a positive effect on satiety, constipation, lipid and glucose metabolism, and gut microbiota (through the prebiotic action). Sweeteners are often used as substitutes for sugar and include polyols and artificial sweeteners [70]. They have different rates of absorption, fermentation, and urinary excretion, so they can impact ketone body production, glycemia, and also gut microbiota in different ways. They are useful in order to increase palatability in KDTs, but their consumption should be limited due to the possible laxative effect.

### 6.4. Beverages

The Mediterranean pattern recommends adequate water intake of at least 1.5–2 L/day in adults [64]. Recommended intakes may vary among people due to age, physical activity, individual characteristics, and weather conditions. Adequate fluid intake is recommended by international consensus and also during KDT treatment [17,24]. In the past, KDT initiation included fluid restriction because it was believed that mild dehydration improved the efficacy of the KD without scientific evidence [17]. Nowadays, most centers no longer restrict fluid intakes in KDTs [17], also because fluid restriction might add to the development of kidney stones [24]. In addition to water, sugar-free herbal infusions and tea may help to complete the requirements, whereas sweetened fruit juices and soft drinks should be consumed in small amounts and set aside for special occasions. These recommendations fit well in KDTs, except for sweetened beverages which are to be avoided completely due to sugar content. Even sugar-free drinks (i.e., cola, orange soda) should be moderately consumed, due to their sweeteners and additives content. The MD recommends wine in moderation, and respecting social beliefs, in the KDTs it must be avoided or limited, given the sugar and alcohol content.

### 6.5. Spices and Herbs

As naturally sugar-free foods and sources of micronutrients and antioxidant compounds promoted in the MD [64], spices and herbs are a good way to vary foods’ flavors and increase dishes’ palatability in KDTs. Moreover, their consumption is recommended in order to reduce salt use [90].

A summary of the recommended frequencies of consumption of the main food groups inspired by the MD pattern is described in Table 3. Portions must be individualized and prescribed by the dieticians/medical doctor or Keto team.

### 6.6. Seasonal, Traditional, Local, Eco-Friendly, and Biodiverse Products

The preference for seasonal, fresh, and minimally processed foods is a key concept in the Mediterranean pattern and can be translated into KDTs. This choice, in fact, allows one to maximize the content of protective nutrients and substances in the diet. Especially in the case of fresh products, several factors influence their nutritional value: the growing methods used, the specific variety chosen, ripeness when harvested, postharvest handling, storage, extent and type of processing, and the distance transported [91]. Unfortunately, fresh foods consumption has been reduced due to the modern lifestyle, encouraging processed foods consumption. However, progress in modern technology minimizes nutrient loss and offers healthy alternatives [92]. The MD strengthens traditional, local, eco-friendly, and biodiverse products, which are to be privileged also in the KDTs. At the basis of the MD, in fact, there is not only a set of foods but also a cultural model incorporating the whole food chain; the way foods are selected, produced, processed, and distributed to the consumers [93]. Thus, the MD is an example of a sustainable pattern [94,95]. Diet sustainability is a concept that integrates a correct food lifestyle with environmentally friendly agricultural production, all with the goal of everyone’s health and well-being at all ages. A “sustainable diet” model is a diet with low environmental impact that contributes to food and nutrition security and to the health of present and future generations. The factors that influence the sustainability of diets are numerous and include not only the type of food present in the diet but also the packaging with which they are marketed, the domestic preparations with the resulting waste, and, in particular for vegetable products, the seasonality and place of cultivation.

All the previously explained recommendations have been summarized in a graphic representation (Figure 1), created as follows: at the base, food items that should sustain the diet and provide the highest energy intake, and at the upper levels, foods to be eaten in moderate amounts or completely avoided. Fat-, protein-, and carbohydrate-rich food sources were represented using three different colors: green for fats, light blue for proteins, and orange for carbohydrates. Within each category, foods were divided into different levels according to their recommended frequency of consumption and serving size. For example, in KDTs, fats represent the main source of energy and therefore are at the base, divided into vegetable fats, placed at the first step of the pyramid because their consumption should be encouraged both in quantity and frequency, and animal fats at the second level because they should be limited. The same applies for other food categories.

## 7. Strengths and Limitations

This narrative review presents some strengths and limitations. First of all, although previous studies reported the use of Mediterranean ketogenic diets, it is the first study in the literature that defines the conceptualization and characteristics of Med–KDTs. Second, it gives graphical and practical recommendations on how to elaborate Med–KDTs. Moreover, the recommendations are thought to be suitable for all types of KDTs.

The current review is a narrative review with a detailed search and a narrative analysis, allowing deep discussion of the topic. However, this approach could have some bias present in all narrative reviews. Due to the specific scope of this study, we did not systematically review all the studies that used a Mediterranean version of high-fat KDTs but focused on those for neurological diseases. Moreover, the proposal is based on products available in Italy and could not necessarily be generalized to other countries. Further studies must be conducted to adapt the Med–KDTs to a wider scenario (Western and non-Western diets). Last, the lack of available relevant references, such as clinical trials and systematic reviews, regarding the use of Med–KDTs and prospective outcomes limits the present findings but incentivizes the creation of new protocols in the field.

## 8. Conclusions

This paper suggests the adoption of a Mediterranean version of ketogenic diets in order to benefit from the multiple protective effects of the MD. This can be summarized as: (i) the preferential use of vegetable fat sources in general, especially olive oil; (ii) the limitation of foods rich in SFA; (iii) the encouragement of HBV protein sources; (iv) the presence of fruit and vegetables at every meal when possible, varying their choices according to seasonality. These suggestions are particularly useful for patients at risk of dyslipidemia. This proposal is based on products available in Italy but may constitute the basis for planning planetary Mediterranean versions of KDTs, taking into consideration local products available in different countries. However, further studies of sufficient rigor and using prospective study designs are needed to prove the benefit of this approach in different health outcomes.

## Figures and Tables

**Figure 1 foods-12-01743-f001:**
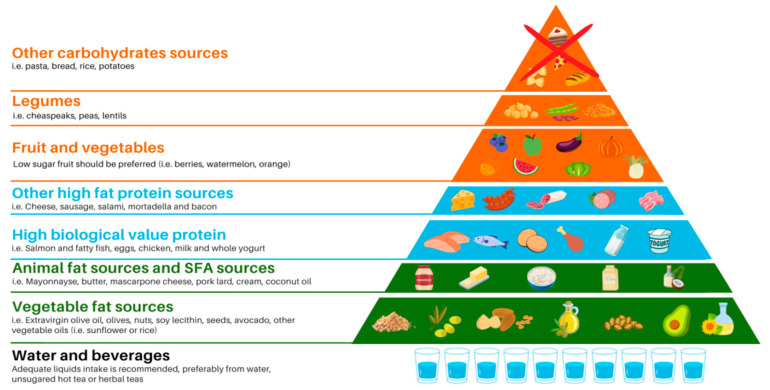
Graphical representation of the practical recommendations for the Med–KDT.

**Table 1 foods-12-01743-t001:** Characteristics of the KDTs.

Type of Diet	Macronutrient Composition			Energy Prescription	Ketogenic Ratio (KR)
	Fat	Protein	Carbohydrates		
CKD [18]	70–90%	7–15% *	3–15%	According to energy requirements	From 1:1 to 4:1
MCT–KD [19]	70% **	10%	20%	According to energy requirements	1:01
MAD [4]	Not measured, fat sources are encouraged	Ad libitum	10 g in children 15 g/day in adolescents 20 g/day in adults	According to energy requirements	1:1–1.5:1
LGIT [20]	60%	30%	40–60 g/day with GI < 50	According to energy requirements	0.6–0.7:1

Recommended intakes are expressed as percentages of the total daily calories and in g/day. CKD intakes are described considering the different ketogenic ratios (from 1:1, with lower fat and higher protein and carbohydrate intakes, to the most restrictive 4:1). * Protein intakes in terms of g/kg are prescribed according to the country’s national recommendation for sex and age. ** MCT up to 40–60% of total daily calories in adults and 10–25% in infants. GI = glycemic index.

**Table 2 foods-12-01743-t002:** Summary of the studies that evaluated the “Mediterranean version” of KDTs for neurological diseases.

Author, Year, Country	Study Design	Study Aim	Population Characteristics				Experimental Diets	Outcomes	Results
			Sample Size and Type of Disease	Mean BMI (kg/m^2^)	Mean Age (Years)	Diet Duration			
Guzel et al., 2019, Turkey	Single-center, prospective study	Efficacy on seizure control and side effects	389 children with drug-resistant epilepsy	Not expressed	6 months to 18 years	3–36 mo	Olive oil-based CKD *Supplements:* sugar-free multivitamin and mineral supplements for all and any liquid medications switched to tablet	At control visits (every 3 mo) - Blood or urinary ketones - Biochemical parameters ** - abdominal ultrasonography and echocardiographic findings	Olive-oil based ketogenic diet effective and well-tolerated; previous history of ACTH use and constipation during KD treatment affect the efficacy of KD treatment. Hyperlipidemia was the most common new-onset adverse effect (50.8%), with an amelioration after diet modification (reducing dietary fats by 20–25%, eliminating saturated fat sources from the diet).
Nagpal et al., 2019 USA	Randomized, double-blind, crossover, single-center pilot trial	Effect on gut microbiome and CSF markers	17 older adults, 11 of them with MCI, and 6 cognitively normal (CN)	Not expressed	64.6 ± 6.4	6 weeks, followed by 6-week washout period	Mediterranean ketogenic diet MMKD (similar to MAD) vs. AHAD: low-fat diet *Supplements:* for all, daily multivitamin and restricted supplements with antioxidant or ketone-inducing effects	Blood ketones weekly Before and after diet interventions: - Gut microbiome - Fecal short-chain fatty acids (SCFAs) - Alzheimer’s disease CSF biomarkers	Several bacteria and SFAs are differently affected by the two diets with distinct patterns between CN and impaired subjects. Specific gut microbial signatures may depict mild cognitive impairment and MMKD can modulate the gut microbiome and metabolites in association with improved AD biomarkers in CSF.
Neth et al., 2020 USA	Pilot Study (Randomized crossover)	Effect on CSF markers, neuroimaging metabolism, and cognition	20 adults with pre-diabetes 11 with SMC and 9 with MCI	28.4 ± 5.7	64.3 ± 6.3	6 weeks, followed by 6-week washout period	Mediterranean ketogenic diet MMKD (similar to MAD) vs. AHAD: low-fat diet *Supplements:* for all, stable physical activity and daily multivitamin supplement	- Alzheimer’s disease CSF biomarkers - Cerebral blood flow and metabolism - Memory performance - Metabolic profile *	MMKD intervention was well-tolerated with good compliance and associated with improved CSF AD biomarker profile, improved peripheral lipid and glucose metabolism, increased cerebral perfusion, and increased cerebral ketone body uptake

* Metabolic profile includes CHOL-t, HDLc, LDLc, TGand glucose (fasting/non-fasting), lipid, lipoprotein. ** Biochemical parameters include CHOL-t, TG, HDL-c, LDL-c, glucose, blood urea, uric acid, creatinine, insulin and adiponectin, interleukin 1 beta (IL-1β), interleukin 6 (IL-6), tumor necrosis factor alpha (TNF-α), interleukin-1 receptor antagonist (IL-1Ra), and interleukin 10 (IL-10). SMC = subjective memory complaint; MCI = mild cognitive impairment CN; mo = months.

**Table 3 foods-12-01743-t003:** Recommended frequency of consumption for Med–KDTs compared to the MD recommendations.

Food	MD Recommendations	Med–KDT Recommendations
Every day consumption		
Olive oil	At every main meal	At every meal (at least lunch and dinner)
Olives, nuts, and seeds	1–2 servings	At least 1 serving
Avocado and soy lecithin	Not mentioned	At least 1 serving
Dairy products	2 servings (preferably low fat)	2 servings (generally high fat)
Mayonnaise, cream, mascarpone cheese, and butter	As little as possible	Maximum 1–2 servings
Fruit	1–2 servings	1–2 servings
Vegetables	>2 servings	>2 servings
Cereals (preferably whole grains)	1 or 2 servings per meal	Not allowed
Water	1.5–2 liters	At least 1.5–2 liters, according to individual needs
Other vegetable and coconut oils	Not mentioned	At every meal in the MCT–KD or CKD plus MCT protocols
Weekly consumption		
Fish	At least 2 portions	At least 2 portions
Eggs	2–4 servings	2–4 servings
White meat	2 servings	2 servings
Red meat	Less than 2 servings	Less than 2 servings
Processed meat	Less than 1 serving	Less than 1 serving
Cheese	Included in the dairy products	2 servings
Legumes	At least 2 portions	At least 2 portions
Occasionally		
Pork lard	As little as possible	As little as possible
Sausages, hamburger, bacon, mortadella	As little as possible	As little as possible
Sugar-free drinks	Not mentioned	As little as possible

## Data Availability

The data generated during the current study are available from the corresponding author upon reasonable request.

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
