# Peer review of "A Review of Ketogenic Dietary Therapies for Epilepsy and Neurological Diseases: A Proposal to Implement an Adapted Model to Include Healthy Mediterranean Products"

_foods, 2023, doi:10.3390/foods12091743_

Round 1

Reviewer 1 Report

The manuscript of Ferraris et al. addresses an interesting proposal about the implementation of a Mediterranean version of Ketogenic dietary therapies for neurological diseases, with potential beneficial effects in patients’ health.

In general, the manuscript meets the requirements established by Foods journal. The abstract is concise with a total of 200 words (maximum limit according to the journal requisites), in a single paragraph briefly explaining the objectives, results and conclusions of the work without headings. The study presents and interesting proposal which is well explained and discussed. The figures and tables are necessary and allow a better understanding of the results. The references are properly presented as they are identified with numbers in square brackets in the text.

However, important major corrections must be done to start considering the publication of this review:

1.      The title must be adjusted to the content of the manuscript by indicating this work is a literature review. A proposal for the new tentative title could be the following: A Review of Ketogenic dietary therapies for epilepsy and neurological diseases: A proposal to implement an adapted model to include healthier Mediterranean products”.

2.      Introduction (Section 1.) must be supported with more relevant references (e.g.: clinical trials and intervention studies supporting the following of Ketogenic dietary therapies in epilepsy, migraine, brain tumors, Alzheimer’s and Parkinson Diseases, and autism spectrum disorder).

3.      For a better understanding, the objectives of this work (page 7) must be included at the end of the Introduction section (page 2).

4.      A section of Materials and Methods must be included in the manuscript to explain the literature review performed by authors.

5.      Section 4 (Mediterranean diet and its effectiveness in CVD prevention) must be greatly summarized and included at the end of Section 3 (Lipid profile and cardiovascular risk during long-term use of KDTs) to not to lose the main focus of the review.

6.      Authors must include a separate section explaining the most important limitations of the work (e.g.: this proposal is based on products available in a specific country, Italy, etc.).

7.      Conclusions section must be enriched to summarize the most important findings and to put in value this work.

Author Response

Dear Reviewer, 

Thank you for the very usefull comments, that surely contributed to improve the manuscript. Attached you can find the report with our comments 

Reviewer 2 Report

The manuscript entitled „Ketogenic dietary therapies for epilepsy and neurological diseases: A proposal to implement an adapted model to include healthier Mediterranean products” is a review article focused on benefits of ketogenic diet in neurological diseases and the possibility to combine ketogenic and Mediterranean diets to obtain more optimized diet. In my opinion, the topic is up-to-date and describes a needed considerations.

I believe that the manuscript fit the scope of the journal and need some improvements.

Section 2: the authors introduced the ketogenic ratio but they do not explain what values are compared. Please specify that, as the readers can be confused.

Please keep the introduced abbreviation and don’t write full names later on.

The script contains many typos, lack of punctuation marks and errors, like “addiction” instead of “addition”, “assumption” instead of “consumption” and so on.

There is a problem with references formatting also.

There is a problem in point 6. All 6.2 and 6.3 are fat sources. Thus point 6.1 should not be distinguished from them. They can be a part of point 6.1.

The manuscript is long and some information are repeated in different sections. It looks like different people wrote different parts, so they introduced some background in their parts. Please reduce the repetitions.

The “eco-friendly” proposition does not refer only to ketogenic diet, but overall. Therefore, in my opinion, it can be omitted as the text is long enough.

Conclusions should not contain references and repeat the information from the introduction. Please stick to conclusions.

Author Response

Dear Reviewer, 

Thank you for the very useful comments, that surely contributed to improve the manuscript. Attanched you can find the report with our comments 

Reviewer 3 Report

Dear Editors/Authors,

The proposed manuscript deals with important and current topic related to ketogenic dieting. The field is very large and I urge Authors to update some of the sections so to improve the quality of the discussion and also to focus it more to the targeted patients as throughout the text they seem to neglect this part of the review and start to recommend the diet in question in a generic way, which should be avoided. See these and other questions and recommendation tracked in the attachment.

Author Response

Dear Reviewer, 

Thank you for the very useful comments, that surely contributed to improve the manuscript. Attached you can find the report with our comments 

Round 2

Reviewer 1 Report

Dear Authors,

After carefully reviewing the last version of the manuscript, I consider this review is worthy of publication.

Sincerely.